# Lack of impact of OCTN1 gene polymorphisms on clinical outcomes of gabapentinoids in Pakistani patients with neuropathic pain

**Abida Shaheen** [1]*, **Syed Mahboob Alam**[2], **Fahad Azam**[1], **Salman Ahmad Saleem**[3], **Moosa Khan**[4], **Syed Saud Hasan**[5], **Afrose Liaquat**[6]

1 Department of Pharmacology & Therapeutics, Shifa College of Medicine, Shifa Tameer-e-Millat University, Islamabad, Pakistan, 2 Department of Pharmacology & Therapeutics, Basic Medical Sciences Institute, JPMC, Karachi, Pakistan, 3 Department of Pain Clinic, Shifa International Hospital, Islamabad, Pakistan, 4 Department of Pharmacology & Therapeutics, Shaheed Zulfiqar Ali Bhutto Medical University, Islamabad, Pakistan, 5 Department of Pharmacology & Therapeutics, Dow Medical College, Dow University of Health Sciences, Karachi, Pakistan, 6 Department of Biochemistry, Shifa College of Medicine, Shifa Tameer-e-Millat University, Islamabad, Pakistan

* abida.scm@stmu.edu.pk

**Data Availability Statement:** All relevant data are within the manuscript and its Supporting information files.

## Abstract

### Background and objective

Gabapentinoids are the first-line drugs for neuropathic pain. These drugs are the substrate of organic cation transporter (OCTN1) for renal excretion and absorption across the intestinal epithelium. Gabapentinoids exhibit wide interindividual variability in daily dosage and therapeutic efficacy which makes titration regimens prolonged for optimal efficacy. The present study aimed to investigate the possible influence of the single nucleotide polymorphism (SNP) of OCTN1 on therapeutic efficacy and safety of gabapentinoids in neuropathic pain patients of the Pakistani population.

### Methods

Four hundred and twenty-six patients were enrolled in the study. All participants were genotyped for OCTN1 rs1050152 and rs3792876 by PCR-RFLP method and followed up for eight weeks. The therapeutic outcomes of gabapentinoids, reduction in pain score, inadequate or complete lack of response, adverse events (AEs) in responders and discontinuation of treatment on account of AEs were recorded for all patients.

### Results

There was no significant association of genotypes and alleles of both SNPs on the clinical response of gabapentinoids ($P > 0.05$). Similarly, significant differences were not found in the reduction of pain scores and AEs among different genotypes in the responders. The present study has reported the association of OCTN1 rs1050152 and rs3792876

**Funding:** This research was supported by Shifa Tameer-e-Millat University, Islamabad [Ref: IRB#514-363-2015]. The funders had no role in study design, data collection and analysis, decision to publish, or preparation of the manuscript.

**Competing interests:** The authors have declared that no competing interests exist.

polymorphisms with clinical outcomes of gabapentinoids for the first time in the real-world clinical setting.

## Conclusion

Our results suggest a lack of influence of OCTN1 genetic variants in the determination of clinical response to gabapentinoids in patients with neuropathic pain in the Pakistani population. These findings signify the role of renal functions in predicting the interindividual variability to therapeutic responsiveness of gabapentinoids.

## Introduction

Pharmacogenetic studies focusing on the genetic variations of membrane transporters involved in the disposition of drugs might explain interindividual variability in the clinical response to substrate drugs [1, 2]. Organic cation transporter 1 protein (OCTN1) is located on chromosome 5q31 and encoded by SLC22A4. It is a multispecific transporter at the apical membrane of the kidney and intestine [3, 4]. The role of nonsynonymous single nucleotide polymorphisms (SNPs) in SL22A4 is implicated in variable absorption and disposition of OCTN1 drugs substrates [2, 5, 6]. Gabapentinoids (gabapentin and pregabalin) are recommended as first-line treatment for neuropathic pain; however, there is wide interindividual variability in dosing requirements, clinical efficacy and tolerability to these drugs which could be attributed to the polymorphisms of genes implicated in the absorption and elimination.

Gabapentin is an OCTN1 pharmacological substrate and the role of a common genetic variant of OCTN1-1507C>T (L503F) has been suggested in the reduced renal tubular secretion of gabapentin [2]. The rs1050152 L503F variant of OCTN1 has been observed with altered metformin uptake signifying the role of OCTN1 genetic variants in the disposition of substrate drugs [5, 7]. However, the association of L503F OCTN1 polymorphism with pharmacokinetics and pharmacodynamics of substrate drugs gabapentin and metformin has been inconsistent and not replicated in different studies [8–10]. Furthermore, the uptake of pregabalin into intestinal epithelial cells has been shown to be mediated by OCTN1 transporter in murine diabetic peripheral neuropathy model [11, 12].

The genetic variants of OCTN1 rs1050152 and intron 1 rs3792876 mutation affecting the binding of runt-related transcription factor 1 protein (RUNX-1) for transcription of SLC22A4 are reported to have a strong association with diabetes mellitus, inflammatory bowel disease, rheumatoid arthritis and autoimmune thyroiditis suggesting the involvement of SLC22A4 in the autoimmune regulatory pathway [13–16]. In addition, increased absorption of inhaled bronchodilator salbutamol with better clinical outcomes has been reported in asthmatic individuals with recessive genotype suggesting the role of rs3792876 homozygous variant in the absorption of OCTN1 drug substrates [17].

Few studies have reported the lack of significant association of genetic polymorphism of OCTN1 on gabapentin population pharmacokinetics [9, 10]. Despite the potential influence of SLC22A4 OCTN1 polymorphism in disruptive gabapentin binding and inflammatory conditions, the association of rs3792876 and rs1050152 with the therapeutic response of OCTN1 substrate drugs has never been studied in real-world clinical settings. With this background, our study aimed to investigate the association between OCTN1 rs1050152 and rs3792876 genetic polymorphisms and efficacy and tolerability of gabapentinoids in neuropathic pain patients of the Pakistani population.

## Materials and methods

### Study design and subjects

The present prospective, observational study was carried out from June 2016 to March 2021 at the Basic Medical Sciences Institute, Karachi and the pain clinic of Shifa International Hospital, Islamabad. The Institutional Review Board & Ethics Committee gave ethical approval of the study protocol (IRB#514-363-2015) following standards of the revised Declaration of Helsinki and the guidelines of Good Clinical Practices.

Four hundred and twenty-six patients with neuropathic pain clinical diagnosis of ≥3 months, aged ≥ 18 years, ≥ 4 pain score on Numeric Rating Scale (NRS), ≥ 40mm score on Visual Analog Scale (VAS) and non-responsive to previous pain medications for more than three months participated in the study. All the participants were recruited in the study after explaining the clinical protocol and obtaining written informed consent. The participants with <60ml/min creatinine clearance or renal impairment, deranged liver enzymes, pregnancy, lactation, anemia, gastrointestinal diseases or using drugs influencing the absorption of the gabapentinoids were excluded from the study.

### Collection of data

The participants receiving different dosages of pregabalin and gabapentin as per standard clinical treatment protocol for neuropathic pain were included in the study. The demographic and clinical details were recorded and patients were followed from baseline and at 2-, 4- and 8 weeks for clinical efficacy and tolerability without any restriction on daily dosage, dosage regimen, weight, gender or ethnicity. NRS was used to measure clinical outcomes at baseline and follow-up visits after the start of gabapentinoids. A clinically significant meaningful improvement of ≥30% (partial responders) and ≥50% (complete responders) in mean pain score over the last seven days with maximum titrated doses was labeled as clinical efficacy [18]. The discontinuation of gabapentinoids due to inadequate or complete absence of response and intolerable adverse events (AEs) were recorded for all patients. The responders were followed for eight weeks to document all adverse events to gabapentinoids.

### OCTN1 rs1050152, rs3792876 genotyping analysis

The genomic DNA extraction procedure using proteinase K and phenol was used for isolation of DNA from EDTA treated whole blood samples. The Fasta sequences of OCTN1 SNPs rs1050152 and rs3792876 were downloaded from dbSNP of NCBI (https://www.ncbi.nlm.nih.gov/snp) and NEBCutter V2.0 program was used for the identification of restriction enzymes (REs) for in silico digestion of sequences and following primers were designed for rs1050152 (Forward: 5'-TCCCAAACTTTCAGGGAAAA-3', Reverse: 5'- CAAGAGTGCCCAGAGAGTCC-3') and for rs3792876 (Forward: 5'-AGGCTAAAGGAGCAGGAAG-3', Reverse: 5'- TCTCAGTGCCTCCCAGAAGT -3').

The polymerase chain reaction (PCR) based restriction fragment length polymorphism (RFLP) was employed for DNA amplification and genotyping. The PCR reaction was performed in a total volume of 25μl consisting of 2μl isolated DNA, 1 μl of each reverse and forward primer (30nmol), 12μl of PCR Master Mix (12.5mM) (5x FIREpol®, Solis BioDyne) and 9μl nuclease-free water in the PCR GeneAmp system (Thermal Cycler Bio RAD T100, USA).

The OCTN1 rs1050152 PCR was carried out at initial denaturing step of 5 minutes at 94˚c, 40 cycles of 30 seconds denaturation at 94˚c, 20 seconds annealing of primers at 55˚c, extension for 30 seconds at 72˚c and then final 3 minutes extension step at 72˚c. PCR conditions for OCTN1 rs3792876 were fixed at the initial step of denaturation at 94˚c for 5 minutes, 35 cycles

at 94˚c for 30 seconds, 60˚c for 30 seconds, 72˚c for 30 seconds and then final step of extension at 72˚c for 3 minutes. The OCTN1 rs1050152 amplified fragment length was 151 base pair (bp) and cleaved by MnlI RE into 123, 102, 28 and 21 base pair products with 21, 28, and 102 bands for C allele and 28 and 123 for T allele on 1% agarose gel (S1A and S1B Fig). The amplified fragment length of OCTN1 rs3792876 was 620 bp and cleaved by HphI into 412, 208, 369, and 43 base pairs with consequent PCR products of 208 and 412 for C allele and 43, 208, and 369 for T allele on 4% agarose gel (S2A-S2C Fig).

## Statistical analysis

Statistical analysis of data was performed using statistical software SPSS program version 23 (SPSS Inc., Chicago, IL, USA). Quantitative data of weight, age, doses of gabapentinoids, pre and post-treatment pain scores were represented as mean ± standard deviation. One-way analysis of variance (ANOVA) with the multiple comparison post hoc Tukey test was applied to calculate intergroup differences among different genotypes. Chi-square or Fisher exact test was used to measure categorical data of adverse events, ethnicity and gender. Binary logistic regression was used to analyze the association of clinical efficacy with OCTN1 genetic variants. A $P$-value < 0.05 was considered to demonstrate a difference of statistical significance. The Hardy-Weinberg equilibrium (HWE) for distribution of genotype frequencies of both polymorphisms was performed using the chi-square ($\chi2$) test. A $P$-value < 0.05 was not considered in concordance with HWE.

## Results

### Patient characteristics

Four hundred and twenty-six patients with neuropathic pain were included in the present prospective observational study. Out of 426, 233 (53.7%) were female and 193 (44.5%) were male. The mean age and weight of all participants were 51.79 ± 9.84 and 73.02 ± 6.15, respectively. Mean serum creatinine and eGFR levels were 0.85 ± 0.16 and 89.37 ± 20.67, respectively. Comparison between baseline and demographic clinical features of non-responders and responders is shown in Table 1. Significant associations between baseline demographic and clinical characteristics among different genotypes of OCTN1 rs1050152 and rs3792876 were not found (S1 and S2 Tables).

### Frequency of OCTN1 polymorphism

Out of 426, four hundred and one patients were available for OCTN1 rs1050152 genotyping and three hundred and ninety-eight patients were available for rs3792876 genotyping. Other patients were excluded for final genotype analysis due to failed genotyping.

The OCTN1 rs1050152 major homozygous CC, heterozygous and minor homozygous CT and TT genotype frequencies were found to be 279 (69.58%) and 122 (30.42%), respectively. The OCTN1 rs3792876 major homozygous CC frequency was 322 (80.90%) whereas heterozygous and minor homozygous CT and TT variants frequency was 76 (19.10%). The genotype distribution of rs1050152 polymorphisms among all subjects did not deviate from HWE ($P = 0.681$); however, the genotype distribution of rs3792876 polymorphism was not consistent with HWE among all subjects ($P < 0.05$). A possible reason for the deviation in rs3792876 SNP could be sampling in the hospital setting in diseased individuals only. Another reason for this deviation could be due to sampling and data collection in tertiary care hospitals in the cities of Rawalpindi and Islamabad that have predominantly Punjabi population.

**Table 1. Demographic and baseline characteristics.**

| Patients (n = 426) | Non-responders (n = 70) | Responders (n = 356) | *P*-value |
|---|---|---|---|
| **Gender** n(%) | | | 0.940 |
| Female | 38 (8.92) | 195 (45.77) | |
| Male | 32 (7.51) | 161 (37.79) | |
| **Age (y)** (Mean±SD) | 51.99±9.38 | 51.75±9.94 | 0.855 |
| **Weight (kg)** (Mean±SD) | 72.74±5.50 | 73.07±6.28 | 0.685 |
| **Ethnicity** n (%) | | | 0.009 |
| Kashmiri | 3 (0.70) | 13 (3.05) | |
| Urdu Speaking | 5 (1.17) | 18 (4.23) | |
| Pathan | 1 (0.23) | 57 (13.38) | |
| Punjabi | 61 (14.32) | 255 (59.86) | |
| Others | 0 (0) | 13 (3.05) | |
| **Serum Creatinine (mg/dl)** (Mean ± SD) | 0.83±0.15 | 0.86±0.16 | 0.124 |
| **eGFR (ml/min/1.73m$^2$)** (Mean ± SD) | 92.70± 21.43 | 88.72±20.48 | 0.141 |
| **Etiology** n(%) | | | 0.520 |
| CPRS | 3 (0.70) | 8 (1.88) | |
| Intercostal neuralgia | 3 (0.70) | 11 (2.58) | |
| Radicular pain | 17 (3.99) | 92 (21.60) | |
| Painful Diabetic neuropathy | 40 (9.39) | 185 (43.43) | |
| Others | 7 (1.64) | 60 (14.08) | |
| **Baseline pain score** (Mean ± SD) | 6.41±0.55 | 7.04±0.66 | 0.001 |
| **Mean pregabalin dose** (Mean ±SD) (285) | 96.11 ±44.58 (45) | 109.48 ±54.99 (240) | 0.125 |
| **Mean gabapentin dose** (Mean ±SD) (141) | 548.00±206.40 (25) | 507.76 ±206.69 (116) | 0.379 |

## Association of OCTN1 polymorphism with clinical efficacy

Based on clinical response to gabapentinoids, 356 (83.57%) patients responded to gabapentinoids whereas 70 (16.43%) were non-responders. The genotype and allele distribution of SLC22A4 rs1050152 and rs3792676 among responders and non-responders is represented in Table 2. Logistic regression analysis revealed that the genotype and allele distribution of OCTN1 rs1050152 and rs3792876 had no significant difference between non-responders and

**Table 2. OCTN1 (rs1050152 and rs3792876) genotype, allelic distribution among responders (complete and partial) and non-responders.**

| rs1050152 (401) (n) | Non-responders (65) n (%) | Responders (336) n (%) | B | *P*-value | OR | (95% CI) |
|---|---|---|---|---|---|---|
| CC (279) | 44 | 235 | Ref categ | | | |
| | (67.69) | (69.94) | | | | |
| CT (108) | 20 (30.77) | 88 (26.19) | 0.194 | 0.514 | 1.214 | 0.678–2.174 |
| TT (14) | 1 (1.54) | 13 (3.87) | -0.890 | 0.397 | 0.411 | 0.052–3.221 |
| C allele | 108 (83.08%) | 558 (83.04%) | | 1.000 | 1.003 | 0.608–1.655 |
| T allele | 22 (16.92%) | 114 (16.96%) | | | | |
| rs3792876 (398) (n) | Non-responders (68) n (%) | Responders (330) n (%) | B | *P*-value | OR | (95% CI) |
| CC (322) | 58 (85.29) | 264 (80.0) | Ref categ | | | |
| CT (60) | 8 (11.76) | 52 (15.76) | -0.356 | 0.381 | 0.700 | 0.316–1.553 |
| TT (16) | 2 (2.94) | 14 (4.24) | -0.430 | 0.576 | 0.650 | 0.144–2.939 |
| C allele | 124 (91.18%) | 580 (87.88%) | | 0.306 | 1.425 | 0.754–2.695 |
| T allele | 12 (8.82%) | 80 (12.12%) | | | | |

B, regression coefficient; P-value, significance level; OR, odds ratio; CI, Confidence interval.

**Table 3. Effect of OCTN1 rs1050152 and rs3792876 polymorphisms on pain score difference in responders (complete and partial) after gabapentinoids treatment.**

| Genotype n (%) | Pain score at baseline (Mean±SD) | Pain score post-treatment (Mean±SD) | Pain score improvement (Mean±SD) | Intergroup differences of genotypes (P-value) | | | One-way ANOVA (P-value) |
|---|---|---|---|---|---|---|---|
| | | | | CC vs TT | CC vs CT | TT vs CT | |
| rs1050152 (n = 336) | | | | | | | |
| CC 235 (69.94%) | 7.02±0.66 | 2.63±1.42 | 4.39±1.29 | 0.995 | 0.725 | 0.912 | 0.734 |
| CT 88 (26.19%) | 7.08±0.65 | 2.69±1.50 | 4.39±1.40 | 0.521 | 0.934 | 0.643 | 0.539 |
| TT 13 (3.87%) | 7.00±0.58 | 3.08±1.32 | 3.92±1.26 | 0.435 | 1.000 | 0.466 | 0.464 |
| rs3792876 (n = 330) | | | | | | | |
| CC 264 (80%) | 7.00±0.65 | 2.69±1.42 | 4.31±1.29 | 1.000 | 0.491 | 0.824 | 0.519 |
| CT 52 (15.76%) | 7.12±0.65 | 2.62±1.43 | 4.50±1.39 | 0.667 | 0.931 | 0.820 | 0.666 |
| TT 14 (4.24%) | 7.00±0.55 | 2.36±1.60 | 4.64±1.34 | 0.624 | 0.606 | 0.930 | 0.445 |

responders ($P > 0.05$). The minor allele frequencies (MAFs) of OCTN1 rs1050152 and rs3792876 (T allele) were 16.92% and 8.82% in non-responders, respectively and exhibited no statistical significance ($P > 0.05$) (Table 2).

For rs1050152, 123 (36.6%) responded partially among 336 responders, whereas for rs3792876 208 responded completely and 122 (36.97%) responded partially to gabapentinoids. For different genotypes of rs1050152 and rs3792876 in the responders' group, significant intergroup differences were not observed after the treatment in terms of pain score (Table 3).

For rs1050152 SNP, a total of sixty-five out of 401 participants discontinued treatment and 41 stopped on account of intolerable adverse events. For rs3792876 SNP, 44 out of sixty-eight participants had unbearable AEs and switched to other drugs. Among responders to gabapentinoids, 138 patients experienced AEs; dizziness and somnolence were the most frequent AEs encountered by 28% of participants. The incidence of AEs was similar among CC, CT and TT genotypes in OCTN1 rs1050152 and rs3792876 variants ($P > 0.05$) (Table 4).

**Table 4. Adverse events (AEs) in patients with OCTN1 rs1050152 and rs3792876 gene polymorphism.**

| AEs | rs1050152 (n = 377) | | | P-value | rs3792876 (n = 374) | | | P-value |
|---|---|---|---|---|---|---|---|---|
| | CC (261) n (%) | CT (102) n (%) | TT (14) n (%) | | CC (301) n (%) | CT (57) n (%) | TT (16) n (%) | |
| Adverse events (+) | 89 (23.61) | 43 (11.41) | 6 (1.59) | 0.501 | 108 (28.88) | 24 (6.42) | 6 (1.60) | 0.720 |
| Adverse events (-) | 172 (45.62) | 59 (15.65) | 8 (2.12) | | 193 (51.60) | 33 (8.82) | 10 (2.67) | |
| Dizziness (+) | 69 (18.30) | 33 (8.75) | 5 (1.33) | 0.608 | 82 (21.93) | 17 (4.55) | 6 (1.60) | 0.696 |
| Dizziness (-) | 192 (50.93) | 69 (18.30) | 9 (2.39) | | 219 (58.56) | 40 (10.70) | 10 (2.67) | |
| Somnolence(+) | 68 (18.04) | 33 (8.75) | 6 (1.59) | 0.398 | 85 (22.73) | 19 (5.08) | 5 (1.34) | 0.750 |
| Somnolence (-) | 193 (51.19) | 69 (18.30) | 8 (2.12) | | 216 (57.75) | 38 (10.16) | 11 (2.94) | |
| Dry mouth (+) | 22 (5.84) | 4 (1.06) | 2 (0.53) | 0.337 | 23 (6.15) | 5 (1.34) | 0 (0) | 0.596 |
| Dry mouth (-) | 239 (63.40) | 98 (25.99) | 12 (3.18) | | 278 (74.33) | 52 (13.90) | 16 (4.28) | |
| Visual blurring(+) | 20 (5.31) | 13 (3.45) | 0 (0) | 0.274 | 26 (6.95) | 5 (1.34) | 1 (0.27) | 0.874 |
| Visual blurring(-) | 241 (63.93) | 89 (23.61) | 14 (3.71) | | 275 (73.53) | 52 (13.90) | 15 (4.01) | |
| Weight gain (+) | 14 (3.71) | 8 (2.12) | 1 (0.27) | 0.798 | 16 (4.28) | 6 (1.60) | 3 (0.80) | 0.102 |
| Weight gain (-) | 247 (65.52) | 94 (24.93) | 13 (3.45) | | 285 (76.20) | 51 (13.64) | 13 (3.48) | |
| Peripheral edema(+) | 25 (6.63) | 12 (3.18) | 1 (0.27) | 0.836 | 28 (7.49) | 6 (1.60) | 1 (0.27) | 0.846 |
| Peripheral edema(-) | 236 (62.60) | 90 (23.87) | 13 (3.45) | | 273 (72.99) | 51 (13.64) | 15 (4.01) | |

## Discussion

Genetic polymorphism of transporter proteins responsible for uptake and disposition of the drugs could significantly modify the pharmacokinetics and clinical response of substrate drugs due to variable therapeutic concentrations. Several physiological and environmental factors such as age, gender, ethnicity, hepatic insufficiency, impaired renal function, concomitant diseases, nutrition, co-administered medications and lifestyle habits could contribute to interindividual differences in the variable clinical efficacy of different drugs [19]. In a recent study, a positive association was found between dose titration and adherence to pregabalin in patients with neuropathic pain. The main factors for nonadherence to the treatment were poor efficacy and tolerability of pregabalin [20]; however, the role of genetic variations of the transporters in relation to the therapeutic efficacy of the substrate drugs needs to be explored.

The pharmacological and pathophysiological roles of common genetic variants of SLC22A4 OCTN1 rs1050152 and rs3792876 in the absorption and disposition of various drugs and autoimmune and inflammatory diseases have been reported in different studies [4–9, 17]. The influence of the OCTN1 genetic variants on the uptake and clearance of substrate drugs might result in variable clinical outcomes. However, there is a scarcity of clinical evidence of the possible association between genetic variants of OCTN1 and the therapeutic outcomes of substrate drugs. The minor allele frequency (MAF) of OCTN1 rs1050152 and rs3792876 were 0.168 and 0.115, respectively. The MAF of L503F rs1050152 in our population is low in comparison to the European and Brazilian population as reported in the 1000 Genome project (www.ncbi.nlm.nih.gov/projects/SNP).

Gabapentinoids are subject to variable interindividual therapeutic responses which could be due to the influence of OCTN1 transporter genetic variants in the absorption and clearance of these drugs. Gabapentinoids are devoid of hepatic metabolism, have negligible binding to plasma proteins and are excreted mainly by filtration in an unchanged form in the urine. The renal elimination of both drugs is linearly associated with creatinine clearance [21–24]. Considering the OCTN1 transporter involvement in the renal secretion of gabapentin, polymorphisms could result in altered pharmacokinetics and clinical response due to increased or decreased transporter activity [2]. Furthermore, different neuropathic inflammatory conditions can modulate the expression and function of different drug-metabolizing enzymes and transporters as a result of the accumulation of inflammatory products [25]. A study investigating common genetic OCTN1 genetic variant (L503F) showed the reduced function of the variant toward gabapentin signifying its role in active tubular secretion [2]. OCTN1 has been implicated in the uptake and altered metformin plasma concentration in OCTN1 L503F genetic variants [5, 7]. A study has recently reported the impact of rs3792876 homozygous genotype in the improved clinical efficacy of inhaled salbutamol suggesting the role of OCTN1 in drug absorption [17]. Our results showed a lack of significant association of OCTN1 rs1050152 and rs3792876 genotypes on the clinical efficacy of gabapentinoids between non-responders and responders ($P = 0.718$ and $0.314$, respectively). Similarly, the T allele frequencies did not demonstrate a significant association between non-responders and responders in both SNPs ($P > 0.05$) (Table 2). According to a recent study, the influence of the L503F OCTN1 1507C>T variant has not been found to be a significant determinant on gabapentin absorption or elimination mechanisms in chronic pain patients. Nevertheless, the association of different genetic variants with the clinical outcomes of gabapentin in chronic pain participants was not explored in the study. Our findings are in agreement with Yamamoto et al. where eGFR was found to be a predictor of gabapentin clearance thus signifying the role of renal function tests in predicting the variability in gabapentin pharmacokinetics [9]. Another study has postulated the role of eGFR as a covariate for total clearance in the accurate

prediction of gabapentin pharmacokinetics irrespective of the glycemic status of patients with diabetic peripheral neuropathy, radiculopathies or different genetic variants of OCTN1 rs1050152 [10]. The findings of our study are also supported by other studies proposing the absence of OCTN1 L503F association with the metformin pharmacokinetics in the Korean and Caucasian population [6, 8].

The OCTN1 expression in the intestinal epithelium and involvement in the absorption process of gabapentinoids has been explored [12]. To the best of our knowledge, any previous study has not investigated the extent of SLC22A4 OCTN1 SNPs influence on the clinical response of gabapentinoids as a consequence of variable absorption and renal disposition of substrate drugs. Since pregabalin and gabapentin elimination in the unchanged form is dependent on renal functions [22–24]; patients with eGFR less than 60 or renal impairment were not included in our study to evaluate the role of OCTN1 SNPs on the therapeutic effectiveness of gabapentinoids. Similarly, a significant difference in pain reduction was not found in responders at baseline and after treatment with pregabalin and gabapentin among different genotypes in both SNPs ($P > 0.05$) (Table 3).

An analysis of the adverse events demonstrated that the majority of the participants experienced dizziness and somnolence as major AEs. The OCTN1 rs1050152 and rs3792876 SNPs were similar among major homozygous CC, heterozygous CT, and minor homozygous TT variants in terms of AEs. The role of L-type amino acid transporter (LAT1) polymorphism has been suggested in the absorption and distribution of gabapentinoids [26]. Our recent study has reported the significant association of LAT1 rs4240803 variant genotype in the variable clinical efficacy and poor tolerability in Pakistani patients with neuropathic pain [27]. To the best of our knowledge, the present study is the first to report the possible association of OCTN1 L503F and rs3792876 on OCTN1 substrates, gabapentin and pregabalin in the South Asian population.

The effect of genetic variants of OCTN1 on the pharmacokinetic parameters of gabapentinoids was not analyzed which is a limitation of our study. However, we excluded the patients with renal impairment to fully understand the clinical relevance of OCTN1 polymorphisms on the therapeutic efficacy of gabapentinoids. Another study taking into account the impact of OCTN1 rs1050152 SNP on absorption and clearance pharmacokinetic parameters of gabapentin could not find any significant association and established renal functions as a main contributor to the clinical pharmacokinetics of gabapentin [9].

## Conclusion

To conclude, the present study did not find a significant contribution of OCTN1 rs1050152 and rs3792876 genetic variants on the therapeutic efficacy and tolerability of gabapentinoids in the Pakistani population with neuropathic pain. This could be due to the low MAF frequency of both SNPs in our population making it imperative to replicate the genetic association studies in different populations in real-world clinical practices.

## Supporting information

**S1 Fig.  A.** OCTN1 rs1050152 polymorphism identified genetic variants of patients on RFLP analysis. **B.** OCTN1 rs1050152 polymorphism identified genetic variants of patients on RFLP analysis.
(TIF)

**S2 Fig.  A.** OCTN1 rs3792876 polymorphism identified genetic variants of patients on RFLP analysis. **B.** OCTN1 rs3792876 polymorphism identified genetic variants of patients on RFLP

analysis. **C.** OCTN1 rs3792876 polymorphism identified genetic variants of patients on RFLP analysis.
(TIF)

**S1 Table. Comparison of baseline demographics and clinical characteristics among different genotypes of OCTN1 rs1050152.**
(DOCX)

**S2 Table. Comparison of baseline demographics and clinical characteristics among different genotypes of OCTN1 rs3792876.**
(DOCX)

**S1 Raw images.**
(PDF)

## Acknowledgments

We acknowledge the study participants for their contribution and commitment to this research project as identifications of SNPs of the relevant genes will facilitate and guide personalized medicine in patients with neuropathic pain.

## Author Contributions

**Conceptualization:** Abida Shaheen, Syed Mahboob Alam, Fahad Azam, Salman Ahmad Saleem, Moosa Khan.

**Data curation:** Abida Shaheen, Syed Mahboob Alam, Fahad Azam, Salman Ahmad Saleem, Afrose Liaquat.

**Formal analysis:** Abida Shaheen, Fahad Azam, Syed Saud Hasan, Afrose Liaquat.

**Methodology:** Abida Shaheen, Syed Mahboob Alam, Fahad Azam, Salman Ahmad Saleem, Moosa Khan, Syed Saud Hasan, Afrose Liaquat.

**Project administration:** Abida Shaheen, Syed Mahboob Alam, Salman Ahmad Saleem.

**Writing – original draft:** Abida Shaheen.

**Writing – review & editing:** Abida Shaheen, Syed Mahboob Alam, Fahad Azam, Salman Ahmad Saleem, Moosa Khan, Syed Saud Hasan, Afrose Liaquat.

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
