## [Decision Letter · Decision Letter 0]

19 Jan 2022

PONE-D-21-32641Lack of impact of OCTN1 gene polymorphisms on clinical outcomes of gabapentinoids in Pakistani patients with neuropathic painPLOS ONE

Dear Dr. Shaheen,

Thank you for submitting your manuscript to PLOS ONE. After careful consideration, we feel that it has merit but does not fully meet PLOS ONE’s publication criteria as it currently stands. Therefore, we invite you to submit a revised version of the manuscript that addresses the points raised during the review process.

We look forward to receiving your revised manuscript.

Kind regards,

Cinzia Ciccacci

Academic Editor

PLOS ONE

Journal Requirements:

Reviewers' comments:

Reviewer's Responses to Questions

**Comments to the Author**

1. Is the manuscript technically sound, and do the data support the conclusions?

Reviewer #1: Yes

2. Has the statistical analysis been performed appropriately and rigorously? 

Reviewer #1: Yes

3. Have the authors made all data underlying the findings in their manuscript fully available?

Reviewer #1: Yes

4. Is the manuscript presented in an intelligible fashion and written in standard English?

Reviewer #1: Yes

5. Review Comments to the Author

Reviewer #1: In the present paper, the authors investigated the possible influence of two OCTN1 SNPs on therapeutic efficacy and safety of gabapentinoids in neuropathic pain patients of the Pakistani population. They did not observe any significant association of genotypes and alleles of both SNPs on the clinical response to gabapentinoids. I think that the strategy followed is correct, the paper is well organized and tables are easily understood, although the authors find no association.

I have a few comments for the authors

1. In the statistical analyzes, the heterozygous and homozygous for the variant allele subjects were merged into a single group, but it would be more complete to report the data of the single genotypic classes.

2. The authors should calculate the Hardy Weinberg equilibrium for the SNPs analyzed.

3. Authors should better explain a possible reasons for the effect of polymorphisms of OCTN1 on gabapentinoids efficacy.

4. The paragraph “PCR product electrophoresis” is superfluous, some of the information is already reported in materials and methods

5. Since the two studied SNPs are located in the same gene, the authors could analyze the frequencies of the haplotypes composed of these two genetic variants.

6. The response to drugs is a multifactorial trait. Other factors, genetic and otherwise, are known to be involved in the response to gabapentinoids? The authors could take them into consideration or at least mention them in the discussion.

7. A revision of the English language is recommended, to correct mistakes and clarify some sentences.

6. PLOS authors have the option to publish the peer review history of their article (what does this mean?). If published, this will include your full peer review and any attached files.

Reviewer #1: No

---

## [Author Response · Author response to Decision Letter 0]

11 Mar 2022

Cinzia Ciccacci

Academic Editor

PLOS ONE

Dear Editor,

We thank the editorial team and reviewers of PLOS ONE journal for their insightful and helpful comments on our manuscript titled "Lack of impact of OCTN1 gene polymorphisms on clinical outcomes of gabapentinoids in Pakistani patients with neuropathic pain" (PONE-D-21-32641). The suggestions of the academic editor and reviewer have been addressed and appropriate changes have been incorporated. The revisions are indicated in bold style in the manuscript and the references section. Following are the responses (written in Italic text) to each of the original comments made by the academic editor and reviewer. 

Response to Academic Editor’s Comments: 

Journal Requirements:

Reply: The manuscript has been revised according to PLOS ONE’s style requirements. The file names have been named according to the instructions.

Reply: The statement about financial disclosure has been included in the cover letter as well as the title page of the manuscript.

Reply: The Data Availability statement has been updated in the cover letter as well as the title page of the manuscript.

Reply: Thank you for highlighting. The references list has been reviewed. We have not cited any retracted article. Some new references have been added in the discussion according to the suggestion of the reviewer and highlighted in the reference list of the manuscript.

Response to Reviewer’s Comments: 

Reviewer #1: In the present paper, the authors investigated the possible influence of two OCTN1 SNPs on therapeutic efficacy and safety of gabapentinoids in neuropathic pain patients of the Pakistani population. They did not observe any significant association of genotypes and alleles of both SNPs on the clinical response to gabapentinoids. I think that the strategy followed is correct, the paper is well organized and tables are easily understood, although the authors find no association.

I have a few comments for the authors.

1. In the statistical analyzes, the heterozygous and homozygous for the variant allele subjects were merged into a single group, but it would be more complete to report the data of the single genotypic classes.

Reply: Thank you for your comment. The details of heterozygous and homozygous for the variant allele subjects have been reported separately in the results tables. 

2. The authors should calculate the Hardy Weinberg equilibrium for the SNPs analyzed.

Reply: Thank you for the suggestion. We have provided the results of HWE analysis in relevant sections of the manuscript.

3. Authors should better explain a possible reasons for the effect of polymorphisms of OCTN1 on gabapentinoids efficacy.

Reply: Thank you for your valuable comment. The possible reasons for OCTN1 polymorphism on the efficacy of gabapentinoids have been added in the discussion segment of the manuscript. 

4. The paragraph “PCR product electrophoresis” is superfluous, some of the information is already reported in materials and methods

Reply: Thank you for highlighting. The redundant lines have been removed from the results and appropriate information has been added in the materials and methods section.

5. Since the two studied SNPs are located in the same gene, the authors could analyze the frequencies of the haplotypes composed of these two genetic variants.

Reply: Thank you for your comment. The haplotype analysis of the two SNPs located in the same genetic region was conducted. There was no significant difference in the frequency of the haplotypes associated with the clinical outcomes so the results have not been included in the manuscript.

6. The response to drugs is a multifactorial trait. Other factors, genetic and otherwise, are known to be involved in the response to gabapentinoids? The authors could take them into consideration or at least mention them in the discussion.

Reply: We thank the reviewer for pointing this out. The suggestion has been addressed in the discussion part of the manuscript. 

7. A revision of the English language is recommended, to correct mistakes and clarify some sentences.

Reply: The revision of English language has been done throughout the manuscript. Some sentences have been revised and shortened for clarity.

Best regards,

Dr. Abida Shaheen (Corresponding author)

---

## [Editor Report · Decision Letter 1]

23 Mar 2022

Lack of impact of OCTN1 gene polymorphisms on clinical outcomes of gabapentinoids in Pakistani patients with neuropathic pain

PONE-D-21-32641R1

Dear Dr. Shaheen,

We’re pleased to inform you that your manuscript has been judged scientifically suitable for publication and will be formally accepted for publication once it meets all outstanding technical requirements.

Kind regards,

Cinzia Ciccacci

Academic Editor

PLOS ONE
---

## [Editor Report · Acceptance letter]

4 May 2022

PONE-D-21-32641R1 

Lack of impact of OCTN1 gene polymorphisms on clinical outcomes of gabapentinoids in Pakistani patients with neuropathic pain 

Dear Dr. Shaheen:

I'm pleased to inform you that your manuscript has been deemed suitable for publication in PLOS ONE. Congratulations! Your manuscript is now with our production department. 

Kind regards, 

on behalf of

Dr. Cinzia Ciccacci 

Academic Editor

PLOS ONE